# Determination of Unsaturated Hydraulic Properties of Seepage Flow Process in Municipal Solid Waste

**Chai Zhang [1,2], Bing Liang [1], Lei Liu [2,3,4,*], Yong Wan [2,3,4] and Qichen Zhu [5]**

1. School of Mechanics and Engineering, Liaoning Technical University, Fuxin 123000, China; zc_cersm@163.com (C.Z.); lbwqx@163.com (B.L.)
2. Institute of Rock and Soil Mechanics, Chinese Academy of Sciences, Wuhan 430071, China; qhdxwy2008@163.com
3. IRSM-CAS/HK PolyU Joint Laboratory on Solid Waste Science, Wuhan 430071, China
4. Hubei Province Key Laboratory of Contaminated Sludge & Soil Science and Engineering, Wuhan 430071, China
5. School of Chemistry and Chemical Engineering, Huazhong University of Science and Technology, Wuhan 430074, China; zqchen@hust.edu.cn
* Correspondence: lliu@whrsm.ac.cn; Tel.: +86-158-7149-7846

**Abstract:** The unsaturated hydraulic characteristics of waste soil are an essential basis for predicting and evaluating leachate migration and distribution in landfills. The saturated water content and permeability coefficient were measured, and a multi-step drainage monitoring experiment was conducted indoors at different dry densities, particle sizes, and degradation ages. Single and dual permeability models were adopted to determine the unsaturated hydraulic characteristic parameters of waste. Results show that dry density and particle size are the key factors affecting the saturated water content and permeability of waste. A single degradation age has little effect on it. Respectively, the saturated water content has a linear relationship with dry density, and permeability has an exponential relationship with dry density under limited experimental data. The overflow numerical inversion method can accurately obtain the unsaturated hydraulic characteristic parameters of wastes and summarizes the values of the unsaturated hydraulic characteristic parameters of wastes with different attributes in the literature and the results of this study. The dual-permeability model performed significantly better than the single-permeability model for water movement, suggesting that a dual-domain description is required for water flow in landfills.

**Keywords:** municipal solid waste; hydraulic properties; preferential flow; numerical inverse

## 1. Introduction

Recirculating leachate in bioreactor landfills and predicting the fate of percolation in a cover system requires understanding and predicting the movement of liquids through municipal solid waste (MSW) landfills. The migration and distribution law of leachate in landfills is predicted by unsaturated flow theory, and the hydraulic characteristics (unsaturated permeability and soil-water characteristics) of the waste required as input when the numerical model is used for simulation are the most critical parameters [1–4].

Some scholars used laboratory experiments to study the changes in the water retention curve of wastes with different densities, degradation ages, and settlement processes [2,4–6], and proposed some single domain models [7,8]. Many reports are published on the use of indoor tests [9–12] and field tests [13–15] on the saturated hydraulic conductivity of waste, but the existing technologies and methods are limited in terms of measuring the unsaturated hydraulic conductivity [16,17]. Thus, the data describing the WRC and $K_\theta$ of MSW are limited [2,9,18–20]. Moreover, using a single domain model to describe the hydraulic characteristics wastes for predicting the water migration inside landfills often has a specific error in engineering practice because of the broad pore characteristics of wastes.

One factor complicating the measurement of refuse constitutive relationships is the prevalence of preferential water flow through landfills, which has been observed in laboratory and field experiments [21,22]. To describe this liquid movement, Gerke et al. [23] established a dual permeability model describing fractured and matrix flows based on the convection–dispersion equation and considered the mass water exchange between the two domains. Using digital image processing technology, the horizontal and vertical slice analysis of waste samples for the dye tracer test also confirms the characteristics of preferential flow in wastes [24]. Although theoretical and experimental investigations prove that the dual-permeability model could reasonably describe the leachate flow in MSW, selecting appropriate hydraulic parameters for the two subdomains is difficult. According to existing data, only a few scholars have determined the dual-domain hydraulic parameters. Han et al. [25] used the dual-permeability model to simulate the leachate flow of injection and desaturation in a 1D waste column through a series of mesoscale experiments. Audebert et al. [26] evaluated the dual-permeation model's hydraulic parameters through an in situ leachate recharge test. However, determining the hydraulic parameters of the dual-permeability model for MSW materials is difficult and thus a research focus.

To compensate for the insufficient research on the unsaturated hydraulic characteristic parameters of real waste and the influence of different physical properties on the changes of the parameters, this study took samples from the Beiyangqiao landfill site in Wuhan and measured the indoor waste's saturated water content and permeability. Through a multi-step drainage experiment and according to the transient data of the monitored outflow, the parameters of the Van Genuchten model (VGM) and the dual permeability model (DPeM) were inversed from the Hydrus-1D. The variation of the unsaturated hydraulic characteristics of wastes with different densities, particle sizes, and degradation ages was analyzed.

## 2. Mathematical Model

### 2.1. Single Permeability Model—Van Genuchten Model

The single permeability model uses the Darcy–Richards equation to describe the transient response of pore water pressure to water migration [27],

$$\frac{\partial \theta}{\partial t} = \frac{\partial}{\partial z}\left[K(h)\frac{\partial h}{\partial z} + K(h)\right] - S \tag{1}$$

where $\theta$ is the water content [$L^3\,L^{-3}$]; $K(h)$ is the unsaturated hydraulic conductivity [$LT^{-1}$]; $h$ is the water head [L]; and $S$ is the source-sink term [-].

The hydraulic properties of wastes can be described by the Mualem-Van Genuchten model [28],

$$S_e = \begin{cases} \frac{1}{[1+(\alpha\bullet|h|)^n]^m} & h < 0 \\ 1 & h \geq 0 \end{cases} \tag{2}$$

$$K(S_e) = K_s S_e^l \left[1 - \left(1 - S_e^{1/m}\right)^m\right]^2 \tag{3}$$

where $S_e$ is the effective saturation [-]; $\alpha$ is the inverse of the air-entry pressure [$L^{-1}$]; $n$ is the pore size distribution index, which determines the slope of the retention curve [-]; $K_s$ is the saturated hydraulic conductivity [$LT^{-1}$]; and $l$ is the tortuosity parameter [-].

### 2.2. Dual Permeability Model

The dual permeability model assumes that the porous media domain is composed of two subdomains, namely, a fissure domain with a large pore size and a matrix domain with a small pore size. The two subdomains are respectively characterized by the Darcy–

Richards equation while describing garbage. The non-equilibrium phenomenon caused by the different pore water velocities in soil has the following governing equations):

$$Fracture : \frac{\partial \theta_f}{\partial t} = \frac{\partial}{\partial z}[K_f(h_f)\frac{\partial h_f}{\partial z} + K_f(h_f)] - \frac{\Gamma_f}{w_f}$$
$$Matrix : \frac{\partial \theta_m}{\partial t} = \frac{\partial}{\partial z}[K_m(h_m)\frac{\partial h_m}{\partial z} + K_m(h_m)] - \frac{\Gamma_f}{1-w_f} \quad (4)$$

where subscripts $f$ and $m$ represent fracture and matrix domains, respectively; $\theta_{f,m}$ is the water volume of the fracture domain (matrix domain) divided by the total volume ($L^3\ L^{-3}$); $K_{f,m}$ is the hydraulic conductivity of the fracture domain (matrix domain) ($LT^{-1}$); $h_{f,m}$ is the head of the fracture domain (matrix domain) (L); $S_{f,m}$ is the source and sink term (-); and $w_f$ is the volume of the fracture domain divided by the total flow domain volume (-). Mass exchange term $\Gamma_w$ is defined as follows:

$$\Gamma_f = \frac{\beta}{a^2}\gamma_w K_a(h_f - h_m) \quad (5)$$

where $\beta$ is a shape factor depending on the geometry of the material (-), $a$ is an effective diffusion path length and the distance from the center of the matrix to the fracture boundary (L), $\gamma_w$ is an empirical scaling factor, and $K_a$ is the effective hydraulic conductivity at the interface between the two domains ($LT^{-1}$).

The dual permeability structure is composed of two subdomains, namely, the fissure and matrix domains, which respectively account for $w_f$ and $w_m$, and variables $\theta$, $h$, and $K$ at any time and space, and are the weighted averages of the two local values:

$$X = w_f X_f + w_m X_m \quad X = h,\ \theta,\ K \quad (6)$$

## 3. Materials and Methods

### 3.1. Experimental Setup

The test device includes three parts: a reactor, the constant head permeation system, and the data acquisition system. That is shown in Figure 1. The reactor has a diameter of 206 mm and a height of 200 mm, and the upper and lower ports are respectively embedded in the cover and base, fixed with screws, and sealed with rubber gaskets. The upper cover plate is screwed into the screw, and a perforated, permeable stainless steel plate is connected to the lower side of the screw to provide different waste compaction densities and prevent the rebound of waste after compaction. Two tensiometers ($-100 \sim 100$ kPa range, $\pm 0.5$ kPa accuracy, Shenzhen Yanzhi Technology Co., Ltd., Shenzhen, China) and moisture content sensors ($0 \sim 100\%$ VWC range, $\pm 2\%$ accuracy, Decagon Devices, Inc., Washington, DC, USA) are installed symmetrically at the height of 7 cm and 14 cm from the upper cover. The base is a truncated cone structure, which facilitates the free flow of water. The constant head infiltration system uses a 2.5 m high steel rod as a pillar, and a pulley suspends a scale bucket. Two valves are at the bottom, using 8.5 L positions in the penetration test. At the water outlet and overflow, the water in the storage tank is pumped to the graduated bucket through the micro-pump to maintain the steady seepage process' constant head gradient. The data acquisition system includes an electronic balance ($20 \sim 4200$ g range, $\pm 0.01$ g accuracy, Ohaus Devices, Inc., New York, NY, USA), a data acquisition instrument, and a computer, which monitors the multi-step drainage outflow, the matrix suction, and the moisture content under different pressure steps through the data acquisition instrument.

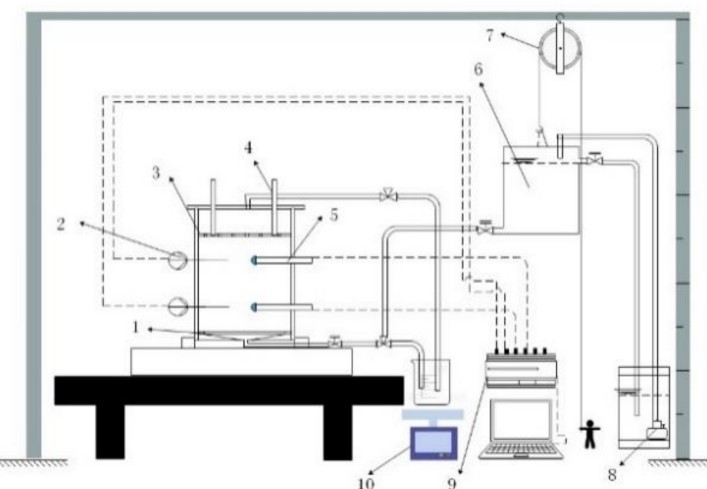

**Figure 1.** Diagram of combined testing system: 1—base, 2—moisture content sensor, 3—porous permeable board, 4—screw, 5—tensiometer, 6—scale bucket, 7—pulley, 8—pump, 9—data acquisition instrument, 10—electronic balance.

### 3.2. Test Materials and Design

The samples used in the test are remolded wastes from the Beiyangqiao landfill in Wuhan. The composition of the waste is shown in Table 1. At 65 °C, the mass change within 2 h is less than 1% of the sample quantity, and the average mass moisture content of the waste is 60% through the drying method. The size of the prepared waste sample should be such that the maximum particle size of the disturbed and undisturbed refuse samples is less than 1/8 of the inner diameter of the analyzer [29], and the dried garbage sample should be cut to the specified size. Firstly, the influence of dry density on the hydraulic characteristics of waste was studied, and the A and B group of waste samples with the same degradation age, same particle size, and different dry density were prepared. Secondly, this study considers the influence of size and age on the parameters of the preferential flow model and adds group C, whose maximum particle size does not exceed 1/4 of the inner diameter, and group D, whose composition is similar to that of aerobic ventilation degradation for one year. The test plan is shown in Table 2.

**Table 1.** Composition of the waste from Beiyangqiao landfill.

| Component | % of Dry Mass |
|---|---|
| Plastic | 23.42 |
| Textiles | 0.95 |
| Wood | 1.80 |
| Rubber | 0.96 |
| Humus | 64.66 |
| Miscellaneous | 8.21 |

**Table 2.** Test scheme for inversion of hydraulic characteristics of waste with different attributes.

| Waste Group | Dry Density (kg m$^{-3}$) | Maximum Particle Size (cm) | Degradation Age (Years) |
|---|---|---|---|
| A | 205 | 2.575 | 13 |
| B | 312.5 | 2.575 | 13 |
| C | 312.5 | 5.15 | 13 |
| D | 205 | 2.575 | 1 |

### 3.3. Determination of Saturated Water Content and Saturated Hydraulic Conductivity

The shredded garbage samples are mixed evenly, divided into five parts, and placed in the reaction kettle by layered compaction. To ensure that the sensor is in close contact with the garbage, a layer of clay is applied on the tensiometer. The sample is saturated with stagnant water. In the initial state, the liquid level of the scale bucket is made to rise slightly above the bottom surface of the sample and then slowly raised at a speed of 1 cm/10 min to remove the air in the sample and achieve complete saturation. The total water injection volume ($V_w$) into the reactor is recorded. Given that the pores are filled with water when saturated, the saturated water content is equal to the porosity:

$$\theta_s = n = \frac{V_W}{V} \tag{7}$$

After saturating the waste, the graduated water bucket's height is raised continuously, and the micropump and the overflow port are opened to ensure the stability of the liquid level of the graduated water bucket at the 8.5 L mark, forming the initial hydraulic gradient ($i$). According to the digital signal tensiometer connected to the reactor ($-100\sim100$ kPa) indicating $h_1$, $h_2$, and the distance $l$ between the two sensors, the hydraulic gradient is $i = (h_1 - h_2)/l$, and at six hydraulic gradients (1.25, 1.5, 1.75, 2, 2.5, 3), the upper spout is connected to the electronic balance to measure the outflow. After the sensor reading stabilizes, the outflow volume ($V$) within $\Delta t$ is recorded. According to Darcy's law, the saturated hydraulic conductivity is

$$K_s = \frac{v}{i} = \frac{V}{iA\Delta t} \tag{8}$$

### 3.4. Multi-Step Drainage Experiment

The multi-step drainage experiment discharges the pore water in the waste in four steps. The four gradients of height reduce the outflow velocity to increase the accuracy of the measurement of the rate data. First, the excess water in the column is drained by adjusting the height of the down tube to be flushed with the top of the sample. Then, four pressure steps are applied in sequence until the height of the down tube is flushed with the bottom of the sample (14, 9, 4, and 0 cm). After completing the drainage test, the four groups of refuse samples from A to D, respectively, flow out 2.32, 1.317, 1.6, and 2.08 L of leachate at 540, 1344, 904, and 450 min. Group B has the longest outflow time and the least outflow. If the multi-step drainage experiment outflow corresponds to the fissure domain's volume [25], then the $w_f$ values of the four groups of refuse samples are 0.418, 0.263, 0.319, and 0.37, respectively.

## 4. Results and Discussion

### 4.1. Saturated Water Content and Saturated Hydraulic Conductivity

The total water inflows of groups A–D were 5.55, 5, 5.02, and 5.57 L, according to the water head saturation method, and their saturated water contents measured by Formula (7) were 0.877, 0.790, 0.793, and 0.880, respectively. Under the same dry density, the saturated water contents were nearly equal, and particle size and degradation age had little effect on it, because under constant dry weight, the pore size and arrangement of different ages and particle sizes varied, but the total pore volume did not vary greatly. However, dry density is the decisive factor that affected the saturated water content. As is shown in the Figure 2, the saturated moisture content of the four groups of garbage samples and the measurement data of Stolz et al. [6] and Breitmeyer et al. [19] have a strong linear correlation with dry density, and for which the functional relation is n = $-7.8183 \times 10^{-4} \rho_d$ + 1.0399.

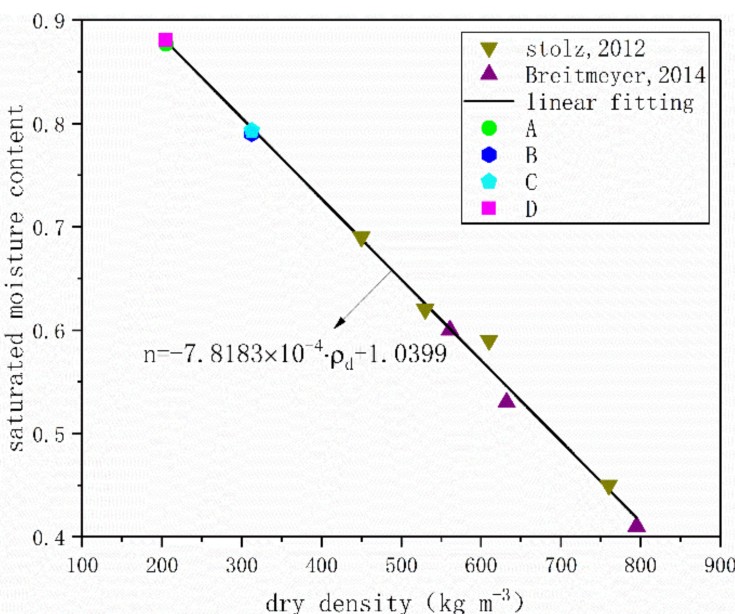

**Figure 2.** Relationship between saturated moisture content and dry density of waste.

This study did not adopt the previous scholars' method of averaging the permeability coefficients corresponding to various hydraulic gradients. Instead, the linear relationship of the permeability rate of waste samples under different hydraulic gradients was analyzed, and the value of the saturated hydraulic conductivity was obtained from the slope of the graph [30]. Figure 3 shows the relationship between the saturated hydraulic conductivity and dry density of the four groups of waste samples. The saturated hydraulic conductivity values of the samples from groups A and B are $4.07 \times 10^{-2}$ and $3.2 \times 10^{-3}$ cm s$^{-1}$, respectively. The permeability of waste decreases with the increase in density. When the density is high, the water migration path is small and the resistance is large, and vice versa. The fitting degree of the saturated hydraulic conductivity values of the four groups of waste samples measured in this experiment with the data of Chen et al. [31] and Beaven et al. [32] is 0.97 on the logarithmic coordinate, for which the functional relation is k = $5.1912 \times 10^{-0.01027\rho d}$. Group C's saturated hydraulic conductivity with the same density and twice the particle size of the group B was $1.44 \times 10^{-2}$ cm s$^{-1}$, which is 4.5 times that of the samples in group B. The increase in particle size causes the refuse skeleton and pore distribution to become more uneven, and the large pore channels guide the preferential flow to increase the permeability. Compared with that of group B, group D's permeability with only one year of degradation has a very small increase, indicating that a single factor (i.e., degradation age) has little effect on the permeability of waste.

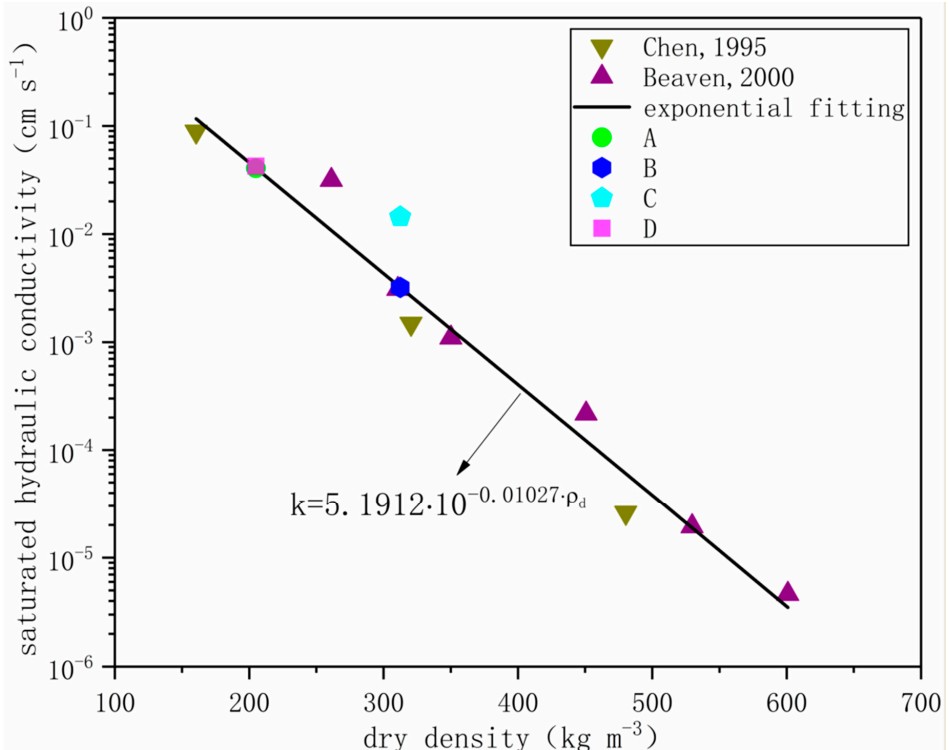

**Figure 3.** Relationship between saturated permeability coefficient and dry density of waste.

*4.2. Hydrus-1D Model Construction*

　　According to the conditions and plans of the multi-step drainage experiment, a 19-cm high waste model was established in Hydrus-1D. The section was divided into uniform materials with 101 nodes. The single and dual permeability models were selected. The initial conditions included constant hydrostatic pressure. For the boundary conditions used for the Hydrus-1D runs, the upper boundary condition flux was set to zero because no water came in/out at the top of the column. The lower boundary condition was set as a variable pressure head, because four pressure steps were applied sequentially by lowering the outflow tubing height in four steps from the top, until the outflow level was in the same level as the bottom of the sample. Owing to the large errors in the monitoring process of the multi-step drainage experiment suction and water content, only the cumulative outflow data was used for the inversion of the VGM model and DPeM model parameters.

　　To improve the certainty and uniqueness of the optimized parameters, the number of unknown parameters must be limited. For the VGM model, the independently measured saturated water content ($\theta_s$) and saturated hydraulic conductivity ($K_s$) are directly used. The tortuosity parameter (l) is assumed to be 0.5, like in the studies of most scholars [2,25,33,34]. For VGM model parameters $\alpha$ and n, residual water content $\theta_r$ is inversed. The DPeM model has 17 parameters, which are nearly three times the number of parameters of the VGM model. Some reasonable assumptions about the model parameters are also made. That is, given the large size of the fracture domain, no other small particles exist, and the water is easily outflowed in the moderate capillary. The fracture domain's residual and saturated water contents are 0 and 1, respectively [25,26,35]. Most of the water in the drainage test is directly discharged from the fracture domain under the action of the drainage experiment, and if fracture domain parameters $\alpha_f$ and $n_f$ are equal to VGM models $\alpha$ and n [35]. Audebert et al. pointed out that the saturated hydraulic conductivity of the fracture domain is 10~103 times that of the single domain $K_s$. Thus, this study assumes

that $K_{sf} = 100\ K_s$; double domain tortuosity parameters $l_f$ and $l_m$ are both 0.5; the saturated water content of the matrix domain can be calculated by the following formula:

$$\theta_{sm} = \frac{\theta_s - w_f \cdot \theta_{sf}}{1 - w_f} \tag{9}$$

Mass exchange coefficients $\beta$, $a$, $\gamma_w$, and $K_a$ are assumed to be 8, 10 cm, 0.4, and $10^{-6}$ cm min$^{-1}$, respectively [23,25,26,33,35,36]. For the matrix domains in the DPeM model, parameters $\theta_{rm}$, $\alpha_m$, $n_m$, and $k_{sm}$ are inversed.

### 4.3. Fitting of Cumulative Outflow in Multi-Step Drainage Experiment

Figure 4 shows that the measured values of the lower boundary cumulative outflow of groups A–D, the VGM model, and the DPeM model, fit to the curve with time. The outflow in the early stage of each pressure step increases linearly with time and then slowly decreases until the outflow reaches equilibrium. The total accumulated outflow and the stable time are negatively correlated with dry density and positively correlated with particle size. The degradation of age has little effect on cumulative outflow, because the dry density of waste is large and the particle size is small. The refuse's internal pores are small, and the water in the refuse is discharged less and more slowly. The longer the degradation age is, the longer the drainage time and the less outflow will be. The comparison of the fitting results indicates that the predicted value and the two models' measured value in the early stage of drainage have certain errors. However, the DPeM model has a better fit than the VGM model, and the coefficient of determination ($R^2$) reaches 0.99 because, in the early stage of the drainage test, the large pores' water flows out preferentially. The fracture domains with large permeability and flow velocity in the DPeM model can just "capture" this dynamic superior flow characteristic. The fit of the VGM and DPeM models can improve as the pore water content of the waste decreases in the later stage of the drainage as well as the permeability. In theory, the DPeM model is more consistent with the pore characteristics of waste, and the errors of the DPeM model are smaller than that of VGM model. The result of numerical inversion is exactly the same. Although more parameters complicate the process of moisture migration, they also improve the accuracy of characterizing the spatial and temporal distribution of leachate.

Figure 5 shows the soil–water characteristic curves of the VGM and DPeM models of the four groups of samples (A–D). The "shapes" of the soil–water characteristic curves of waste with different attributes have apparent differences. The four groups of samples using the DPeM and VGM models have intersection points, which intersect at water contents of approximately 0.7 and 0.6, respectively, and the soil–water characteristic curves of the same dry density samples tend to overlap at the later stage of the water content intersection point. This phenomenon indicates that when the sample is near saturation, the soil–water characteristics of waste are only related to the compaction density, but particle size and age have little effect on soil–water characteristics of waste. This result is consistent with that of Han et al. [25], who pointed out that the degree of waste compaction significantly influences the relationship between matrix suction and volumetric water content. In the pre-saturation stage, as the dry density (A, B) particle size (B, C) increases and the degradation age (A, D) decreases, the soil–water characteristic curve of waste becomes more "steeply standing" because the pore ratio of the refuse decreases, the difficulty of draining the internal pore water increases, and the dehumidification decelerates. Under the same suction, the water content of the DPeM model is lower than that of the VGM model. The DPeM model exhibits a more substantial water release than the VGM model, which is consistent with the migration characteristics of the preferential flow of macropore water.

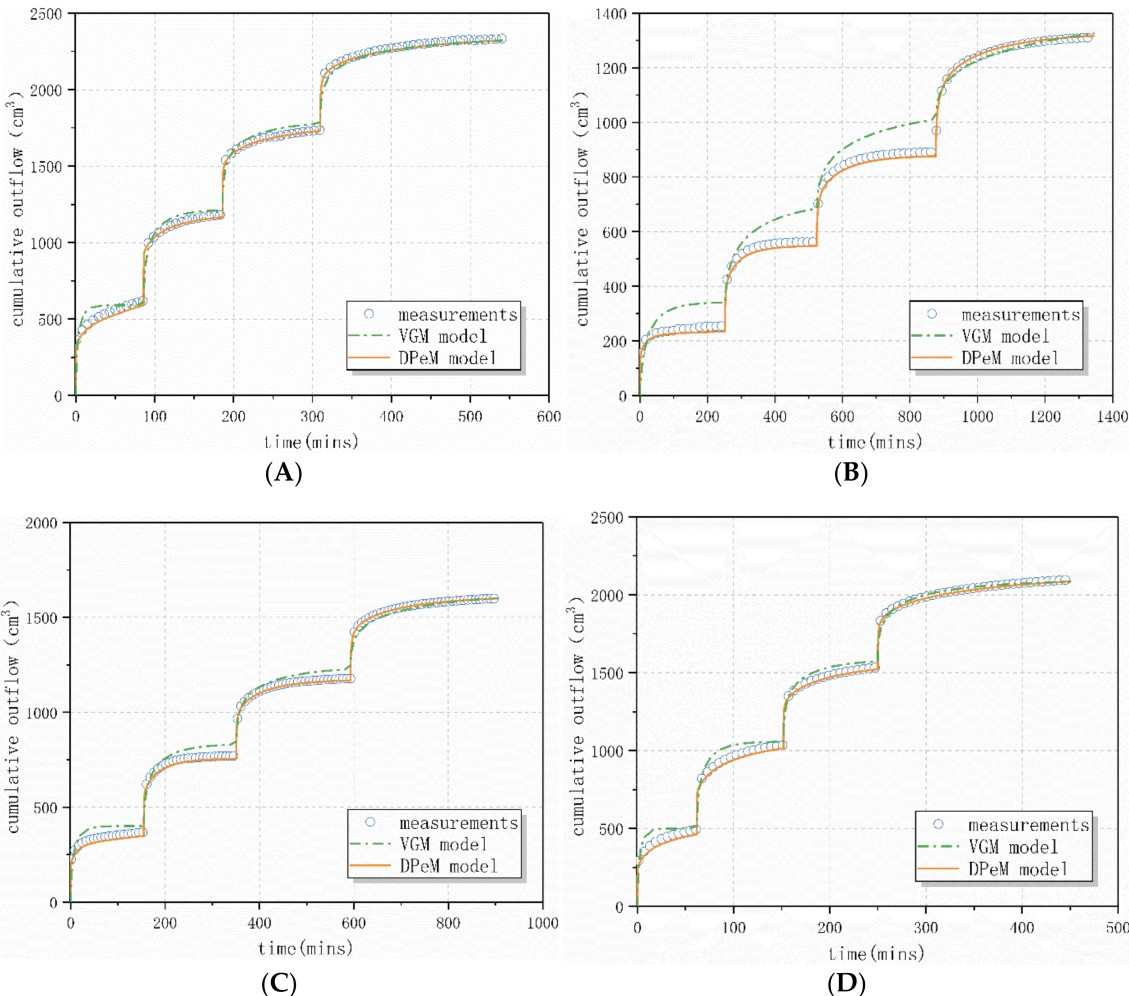

**Figure 4.** Cumulative outflow data for groups (**A–D**) during drainage experiments with inverse modeling fits using best-fit parameters.

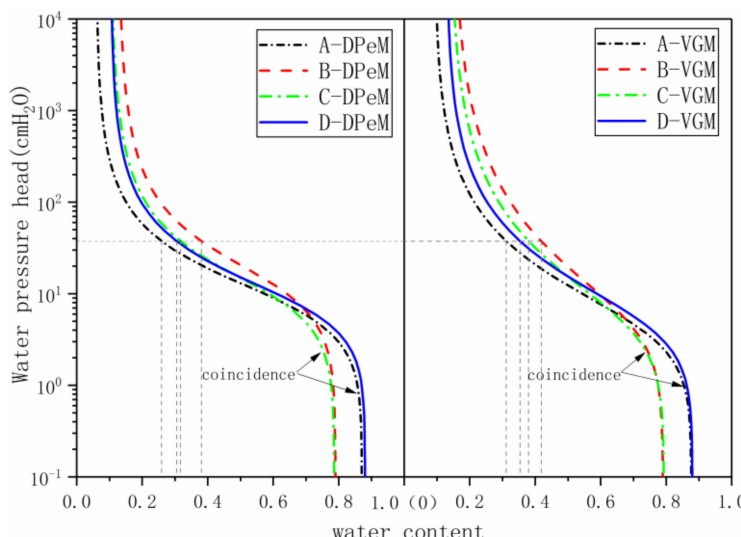

**Figure 5.** Characteristics of soil water in different physical properties.

Figure 6 shows the curves of the four (A–D) samples' unsaturated hydraulic conductivity and water content in the VGM and DPeM models. The permeability increases

with the volumetric water content. The unsaturated hydraulic conductivity values of the four groups of samples in the DPeM and VGM models vary over 13 and 11 orders of magnitude, respectively, and the permeability of the DPeM model is always greater than that of the VGM model. As the dry density increases, the permeability function curve "shifts" to the right, and the water content and permeability change range and the peak hydraulic conductivity decrease. Given that the difference between samples B and C lies in particle size, these two groups of fracture domains should be different, and the water in the matrix domain of the refuse pores should be nearly the same. The graph shows that the DPeM permeability function curve of samples B and C at a low water content (matrix Domains) are nearly coincidental and fractal at a high water content, matching the characteristics of low water content matrix flow and high-water content fractured flow. The VGM model of pore homogenization cannot reflect this flow behavior. The A and D samples of different degradation ages are more permeable than D at $\theta \leq 0.4$, and the permeability functions nearly overlap at $\theta \geq 0.4$, because when the density and particle size are the same, group D (with a shorter degradation age than the other groups) will contain certain organic components, and the residual moisture content will be higher than that of group A. At this time, the proportion of matrix domains in group D ($w_m = 0.63$) will be higher. In group A ($w_m = 0.592$), for the same average volumetric water content, the volumetric water content of the matrix domains of group D will be significantly less than that of the group A matrix domains.

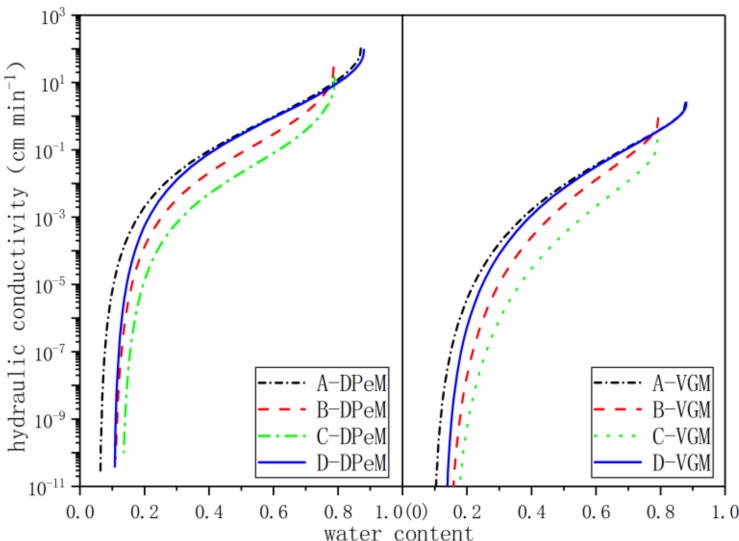

**Figure 6.** Waste hydraulic conductivity and moisture content curve of different properties.

### 4.4. Analysis of Influencing Factors of Hydraulic Characteristic Parameters

#### 4.4.1. VGM Model Parameter Analysis

　　Table 3 shows the hydraulic characteristic parameters inversed by the VGM model in the multi-step drainage experiment on groups A–D. Different dry densities, particle sizes, and degradation ages impact the hydraulic characteristics of waste. Group A's air entry pressure value and residual moisture content are the smallest, and those of group B are the largest. The VGM model parameters (a and n) of the four groups of waste samples vary from 0.181 to 0.21 and 1.45 to 1.618, respectively, and the residual moisture content varies from 0.092 to 0.147, values which are similar to those in the studies of Zardava [35] and Breitmeyer et al. [19], that is, $\alpha \in [0.047, 0.355]$ and $n \in [1.25, 2.6]$ are in the same range but smaller than the proposed residual moisture content (i.e., $\theta_r \in [0.15, 0.34]$). The reason for the small value is that this test uses stale garbage samples with a degradation age of up to 13 years. Microbial degradation significantly reduces the organic component and content, reducing the residual moisture.

**Table 3.** Hydraulic parameters of waste VGM model with different properties (±values are 95% confidence interval).

| Group | $\theta_r$ * | $\theta_s$ | $\alpha$ * $(cm^{-1})$ | n * | $k_s$ (cm $min^{-1}$) | l | $R^2$ |
|---|---|---|---|---|---|---|---|
| A | $0.092 \pm 0.0012$ | 0.877 | $0.210 \pm 0.034$ | $1.610 \pm 0.029$ | 2.442 | 0.5 | 0.9852 |
| B | $0.147 \pm 0.052$ | 0.790 | $0.173 \pm 0.059$ | $1.45 \pm 0.061$ | 0.192 | 0.5 | 0.9759 |
| C | $0.139 \pm 0.046$ | 0.793 | $0.189 \pm 0.037$ | $1.50 \pm 0.013$ | 0.864 | 0.5 | 0.9824 |
| D | $0.128 \pm 0.091$ | 0.880 | $0.181 \pm 0.015$ | $1.618 \pm 0.081$ | 2.55 | 0.5 | 0.9871 |

* Fitting parameters.

The comparison of the parameters of groups B and C (same dry density, degradation age) of different particle sizes shows that the larger the particle size is, the larger parameters $\alpha$ and n, and the smaller the residual moisture content are because when the particle size of the material increases, the pores inside the waste become non-uniformly distributed, and the number of large pores increases. Dewetting can occur with very little suction (air-entry pressure decrease), and moisture retention weakens (residual moisture content decreases) under a small suction force, which is consistent with the variation law of hydraulic parameters inversed by Han et al. [25] with a uniform paper material and different particle size scales. For Group D, with a degradation age of one year compared to Group A with a degradation age of 13 years, parameter $\alpha$ is 0.029 $cm^{-1}$ smaller, the residual water content is 0.036 larger, and pore size distribution index n is basically the same. The higher the degradation age is, the higher the air entry pressure and the lower the residual moisture content are, indicating that the microbial degradation reaction consumes the rich organic components of the waste, increasing the number of fine particles and reducing the moisture retention.

With the dry density increases, VGM model parameters $\alpha$ and n decrease and the residual moisture content increases significantly. The reason is that increasing the compaction density will reduce the porosity and increase the air entry pressure of the waste. The original large pores will be converted into small pores (increasing the number of small pores) to make the pore distribution more uniform. The leachate stored in the small pores is rarely discharged or cannot be discharged under gravity, and the residual water content increases. Figure 7 summarizes the parameters of the soil–water characteristics from the studies of domestic and foreign scholars. This study provides the parameters of the soil–water characteristics of shallow waste (dry density between 200–400 kg $m^{-3}$), which are within the confidence range. The values of $\alpha$ tend to decrease, corresponding to an increase in the air entry suction of MSW, as the dry unit weight of MSW increased in the laboratory specimens. This phenomenon suggests that increases in dry unit weight results in increases in air entry suction, which may occur because of the compression of the largest pores in MSW that governs the air entry suction.

### 4.4.2. DPeM Model Parameter Analysis

Table 4 shows the multi-step drainage experiment on the groups A–D using the DPeM model inversion and the hydraulic characteristic parameters proposed by related scholars. The matrix domain parameters ($\alpha_m$ and $n_m$) of the four groups of refuse samples vary from 0.067 to 0.102 and 2.03 to 2.15, respectively. Residual moisture content $\theta_r$ varies from 0.102 to 0.171, and the air entry pressure and residual moisture content of the B waste matrix domain are the largest. The comparison of the parameters of the DPeM and VGM models under the assumption that fracture domains $\alpha_f$, $n_f$, and $k_{sf}$ are respectively equal to VGM model $\alpha$, n, and 100 k, the residual water content of matrix domain $\theta_{rm}$ is on average 13.7% larger than that of VGM model $\theta_r$, because the matrix domain mainly has small pores; the refuse skeleton and pores are evenly distributed; the leachate stored is difficult to release; and the moisture retention is excellent. The value of the residual water content of the matrix domain is similar to that in the study of Audebert et al. [26] by 10.2%. Matrix domains $\alpha_m$ and $n_m$ are approximately 0.46 and 1.36 times VGM models $\alpha$ and n, indicating that the matrix domain has a uniform distribution of small pores and a sizeable

air entry pressure, which is much smaller than that proposed by Han et al. and Audebert et al. (i.e., $\alpha_m \approx 0.01 \alpha$). The particle size of stale waste is speculated to be smaller, and the fitting parameters are not unique. Further experimental research and demonstration are needed. The pore size distribution index of matrix domain $n_m$ is 1.29 times that of VGM model n, which is similar to that in this study.

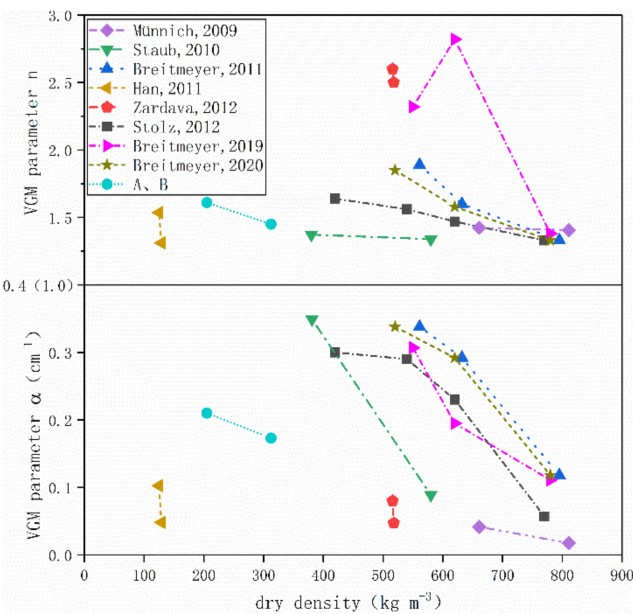

**Figure 7.** Relationship between VGM model parameters $\alpha$, n and dry density.

**Table 4.** Hydraulic parameters of waste DPeM model with different properties (±values are 95% confidence interval).

| Matrix | $\theta_{rm}$ * | $\theta_{sm}$ | $\alpha_m$ * (cm$^{-1}$) | $n_m$ * | $k_{sm}$ * (cm min$^{-1}$) | $l_m$ |
|---|---|---|---|---|---|---|
| A | 0.102 ± 0.064 | 0.778 | 0.098 ± 0.0025 | 2.15 ± 0.371 | 0.198 ± 0.073 | 0.5 |
| B | 0.171 ± 0.048 | 0.715 | 0.067 ± 0.0097 | 2.03 ± 0.539 | 0.072 ± 0.013 | 0.5 |
| C | 0.152 ± 0.087 | 0.686 | 0.084 ± 0.0049 | 2.12 ± 0.754 | 0.133 ± 0.022 | 0.5 |
| D | 0.165 ± 0.055 | 0.81 | 0.102 ± 0.0035 | 2.10 ± 0.215 | 0.203 ± 0.081 | 0.5 |
| [25,26,35] | 0.15~0.22 | 0.65~0.83 | 0.0015~0.25 | 1.5~2.5 | 0.003~0.6 | 0.5 |
| **Fracture** | $\theta_{rf}$ | $\theta_{sf}$ | $\alpha_f$ (cm$^{-1}$) | $n_f$ | $k_{sf}$ (cm min$^{-1}$) | $l_f$ |
| A | 0 | 1 | 0.21 | 1.62 | 244.2 | 0.5 |
| B | 0 | 1 | 0.173 | 1.45 | 19.2 | 0.5 |
| C | 0 | 1 | 0.189 | 1.50 | 86.4 | 0.5 |
| D | 0 | 1 | 0.181 | 1.618 | 255 | 0.5 |
| [25,26,35] | 0 | 1 | 0.08~0.72 | 1.5~2.0 | 6~178.55 | 0.5 |
| **Transfer Term** | $w_f$ | $\beta$ | $\gamma_w$ | a (cm) | $k_a$ (cm min$^{-1}$) | $R^2$ |
| A | 0.418 | 8 | 0.4 | 10 | $10^{-6}$ | 0.9931 |
| B | 0.263 | 8 | 0.4 | 10 | $10^{-6}$ | 0.9963 |
| C | 0.319 | 8 | 0.4 | 10 | $10^{-6}$ | 0.9984 |
| D | 0.37 | 8 | 0.4 | 10 | $10^{-6}$ | 0.9976 |
| [25,26,35] | 0.1~0.657 | 3~15 | 0.4 | 2.5~10 | $10^{-6}$~$10^{-1}$ | |

* Fitting parameters.

The analysis of the parameters' influencing factors shows that the influence of dry density, particle size, and degradation age on the hydraulic characteristics of the DPeM model matrix and fracture domains is the same as that of the VGM model. Matrix domains (fracture domain) $\alpha_{m,f}$, $n_m$, and $_f$ are negatively correlated with dry density and degradation age and positively correlated with particle size. Residual moisture content

$\theta_{rm}$ is positively correlated with dry density and particle size and negatively correlated with degradation age.

### 5. Conclusions

In this study, the saturated water content and hydraulic conductivity of wastes with different physical properties were measured indoors based on the overflow and numerical inversion methods. A multi-step drainage experiment was carried out. The single and double permeability models were inversed using the cumulative outflow value. The parameters and influencing factors of the hydraulic properties of refuse soil at different dry densities, particle sizes, and degradation ages were studied. The following conclusions are drawn:

(1) Dry density and particle size are the key factors that affect the saturated moisture content and hydraulic conductivity of wastes, while degradation age has little effect on them. The saturated moisture content has a linear relationship with dry density, and hydraulic conductivity has an exponential relationship with dry density under limited experimental data.

(2) The numerical inversion of the hydraulic characteristic parameters through the overflow method has the advantages of simplicity and efficiency. The hydraulic characteristics of stale waste with a dry density of 200–400 kg m$^{-3}$ are as follows: When using the VGM model for characterization, the $\alpha$ and n change ranges are 0.181–0.21 cm$^{-1}$ and 1.45–1.618, respectively, and the residual moisture content $\theta_r$ change range is 0.092–0.147. When using the DPeM model for characterization, the parameters of matrix domains $\theta_{rm}$ and $n_m$ can be approximated to be 1.16 and 1.36 times those of $\alpha$ and n in the VGM model, respectively. The reciprocal of air intake value $\alpha_m$ and $K_{sm}$ must be studied and verified further.

(3) Dry density, particle size, and degradation age all impact the hydraulic characteristics of waste; that is, the more significant the dry density is, the lower the reciprocal of air entry pressure $\alpha$ and pore size distribution index n are, and the higher the residual moisture content $\theta_r$ will be. The influence of particle size on the hydraulic characteristics is opposite to that of dry density. The larger the degradation age is, the smaller are the reciprocal of air entry pressure $\alpha$, pore size distribution index n, and residual moisture content $\theta_r$.

(4) The dual permeability model fits the free drainage process of bodyweight force in the indoor model more accurately than the single permeability model, proving the preferential permeability of water in waste and indicating engineering significance for predicting the water migration in landfills.

**Author Contributions:** The work was conducted by L.L., B.L., Q.Z. and Y.W.; this paper was written by C.Z., who reviewed and improved the manuscript with comments; the data compilation and statistical analyses were completed by all authors. All authors have read and agreed to the published version of the manuscript.

**Funding:** This research was funded by the National Natural Science Foundation of China (41977254, 41774197), the Foundation for Innovative Research Groups of Hubei Province (2019CFA012), Wuhan science and technology conversion special project (2018060403011348) and the Youth Innovation Promotion Association CAS (2017376).

**Data Availability Statement:** Not applicable.

**Conflicts of Interest:** The authors declare no conflict of interest.

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
