# Peer review of "Determination of Unsaturated Hydraulic Properties of Seepage Flow Process in Municipal Solid Waste"

_water, doi:10.3390/w13081059_

Round 1

Reviewer 1 Report

My major comment is about model calibration. This section requires significant modifications and deeper discussions concerning:

  • The DPeM model allows a better fit compared to the VGM model. This may be a consequence of the significant difference in the number of parameters for both models. The comparison should be improved using model discrimination criteria.
  • Effects of some assumptions (L224-225, 227-228) should be better analyzed. The inverse procedure could be re-run by using a factor 10 or 1000 between Ks and Ksf for example and provide different results. The dependence of these results on these assumptions should be discussed.
  • The parameter estimation should include parameter uncertainties related to calibration. The difference for parameters a (0.181 to 0.21) or n (1.45 to 1.618) may be not significant. Same comment for the comparison of parameters between groups. Same comments also for the comparison between DPeM model and VGM model.

Concerning the experiments:

  • More details should be given on the equipment (tensiometers, moisture sensors, …).
  • Experiment reliability (or measurements errors) should be quantified. It is important information for parameter estimation through model calibration.

Finally, I wonder if Darcy’s model is valid especially for group C where the particle size is quite big compared to the bucket size. This should be at least discussed in the paper.

Author Response

Dear Reviewer

Thank you for giving us the opportunity to revise our manuscript “Determination of unsaturated hydraulic properties of seepage flow process in municipal solid waste”, Ref. No.: Water-1158616. We are grateful for the valuable and constructive comments from reviewers which have inspired us to enrich the content of the paper and to refine on its clarity. Efforts have also made to improve the use of English for expression of our work and results. We use the "Track Changes" function in Microsoft Word to mark all the changes in the revised manuscript.

We hope that you now find the manuscript suitable for publication in Water.

Thank you for your kind consideration of this manuscript.Detailed comments are attached in WORD file.

Yours sincerely,

Chai Zhang

Reviewer 2 Report

Table 2. The waste groups A and B are not described in the text.

The table needs a minor corrected.

The letter L is used as the distance between the tensiometers and also as a unit of water volume.

In the reviewer's opinion, the equations shown in Figures 2 and 3 should also be written in the text.

Figure 4 caption should be completed with an explanation of what is presented in Figures a, b, c and d.

Author Response

(The authors gave the same response as above.)
